# Neurophysiological Evaluation of Students’ Experience during Remote and Face-to-Face Lessons: A Case Study at Driving School

**DOI:** 10.3390/brainsci13010095

**Published:** 2023-01-03

**Authors:** Ilaria Simonetti, Luca Tamborra, Andrea Giorgi, Vincenzo Ronca, Alessia Vozzi, Pietro Aricò, Gianluca Borghini, Nicolina Sciaraffa, Arianna Trettel, Fabio Babiloni, Manuel Picardi, Gianluca Di Flumeri

**Affiliations:** 1Department of Anatomical, Histological, Forensic and Orthopedic Sciences, Sapienza University of Rome, 00161 Rome, Italy; 2BrainSigns srl, 00198 Rome, Italy; 3Laboratory of Industrial Neuroscience, Department of Molecular Medicine, Sapienza University of Rome, 00161 Rome, Italy; 4School of Computer Science and Technology, Hangzhou Dianzi University, Hangzhou 310018, China; 5IlTergicristallo.it, c/o UNASCA, 00194 Rome, Italy

**Keywords:** face-to-face education, remote education, learning performance, brain activity, heart activity, skin conductance, neurophysiological approach, wearable devices

## Abstract

Nowadays, fostered by technological progress and contextual circumstances such as the economic crisis and pandemic restrictions, remote education is experiencing growing deployment. However, this growth has generated widespread doubts about the actual effectiveness of remote/online learning compared to face-to-face education. The present study was aimed at comparing face-to-face and remote education through a multimodal neurophysiological approach. It involved forty students at a driving school, in a real classroom, experiencing both modalities. Wearable devices to measure brain, ocular, heart and sweating activities were employed in order to analyse the students’ neurophysiological signals to obtain insights into the cognitive dimension. In particular, four parameters were considered: the Eye Blink Rate, the Heart Rate and its Variability and the Skin Conductance Level. In addition, the students filled out a questionnaire at the end to obtain an explicit measure of their learning performance. Data analysis showed higher cognitive activity, in terms of attention and mental engagement, in the in-presence setting compared to the remote modality. On the other hand, students in the remote class felt more stressed, particularly during the first part of the lesson. The analysis of questionnaires demonstrated worse performance for the remote group, thus suggesting a common “disengaging” behaviour when attending remote courses, thus undermining their effectiveness. In conclusion, neuroscientific tools could help to obtain insights into mental concerns, often “blind”, such as decreasing attention and increasing stress, as well as their dynamics during the lesson itself, thus allowing the definition of proper countermeasures to emerging issues when introducing new practices into daily life.

## 1. Introduction

Fully online or blended education started in the 1990s with the advent of the Internet and the World Wide Web, immediately generating favourable impacts for people who live in remote locations or who want to save travel time and costs [1]. With the increasing progress of information and communication technologies, online education has become more technologically, economically, and operationally feasible. The possibility of providing educational programmes expanding one’s catchment area and, at the same time, saving costs related to physical facilities and personnel has fostered the establishment of companies dedicated exclusively to online education and has also caused institutes and organisations traditionally based on face-to-face education to begin evaluating the possibility of delivering at least hybrid modalities. In 2016, Dziuban and colleagues [2] described the evolution of online education in four phases, primarily using the USA context: the 1990s (Internet-propelled distance education), 2000–2007 (increasing use of Learning Management Systems—LMSs), 2008–2012 (growth of Massive Open Online Courses—MOOCs), and beyond, with the growth of online higher education enrolments outpacing traditional higher education enrolments. A few years ago, a report by the Association to Advance Collegiate Schools of Business (AACSB), based on data collected from 521 accredited schools representing 36 countries, showed an increase in the number of schools offering fully online degree programmes at all levels [3]. According to the report, the proportion of schools offering online degrees increased from 25 to 37 percent in the previous five years [4]. However, this growth has been dampened by widespread doubts and misgivings about the actual effectiveness of remote/online learning compared to face-to-face education. Several reports over the years have shown that, for the past decade, faculty perceptions towards technology and online education have not changed much and remained negative [5].

Due to the recent historical scenario, emerging from the COVID-19 pandemic crisis that imposed major restrictions and forced confinement in daily life, alternative solutions to normal activities that required the presence of people were found. One of the sectors most affected by this necessary change has been education and training, which has inevitably forced the use of online/remote learning modes. 

While dealing with the implementation of online education in educational programmes, the major challenge has been identified in retaining social contacts and maintaining ongoing learning [6]. Online education has become an effective means to provide educational activities and prevent the possible loss of academic sessions because of the prolonged lockdown. However, research on online education shows that students displayed a wide range of reactions and behaviours, with most expressing anxiety toward online learning, less motivation and engagement, disappointment regarding graduation ceremonies and general dissatisfaction because of the perception of the difference between online and standard in-class learning [7].

In this scenario, the scientific community started employing new tools and methodologies provided by neuroscientific disciplines to assess students’ experiences from a psychological and cognitive point of view and, therefore, to investigate the differences between the different modalities [8]. The rationale is to employ neuromonitoring devices, i.e., systems for recording students’ brain activity (electroencephalography (EEG) or functional Near-Infrared spectroscopy (fNIRs), heart activity (photopletismographic (PPG) or electrocardiographic (ECG) signals) and skin sweating (electrodermal activity (EDA)), and to obtain aggregate and synthetic indicators, i.e., neurometrics [9,10,11], of cognitive phenomena relevant for the field. A lot of works have been published on this topic, but without tackling the matter in a comprehensive way. For example, the majority of works considered only one modality, i.e., in-person [12] or remote, and, in turn, with online interactive streaming or pre-recorded non-interactive courses [13,14,15,16]. On the other hand, these works usually focused on one specific cognitive and mental state, mainly attention and/or concentration [17,18,19], cognitive load or workload [20,21,22], engagement [23,24] or different affective states (frustration, meditation, etc.) [25,26,27]. Finally, each work usually employed just one technology, i.e., EEG or fNIRs or EDA or Eye-Tracking, instead of trying to obtain a multimodal perspective of the user’s experience. 

Until a few years ago, this concern was mainly due to the invasiveness of such systems, the restrictions they imposed (for this reason, the majority of the studies were not in realistic settings but in a laboratory) and their interference with the user’s natural behaviour, but recent technological progress and the establishment of wearable sensing technology [28,29,30,31] now enable new opportunities for conducting less invasive studies in more ecological settings.

The present study takes advantage of the last considerations. Wearable devices to measure EEG, PPG and EDA signals were employed on forty students at a driving school, in a real classroom, in order to neurophysiologically evaluate their learning experiences in both face-to-face and remote conditions. The use of neuroscientific methodologies should guarantee a deeper and more objective evaluation of students’ experiences, being based on the direct analysis of mental and physiological reactions [32].

The field of driving training was chosen because it is also experiencing a boom of fully online driving schools providing courses for theory exams, both in the European dimension and worldwide. In the context of driving education, there is the perception that some specific topics, such as those related to road safety, are particularly relevant, and therefore, “in-presence” (face-to-face) education should be encouraged in any case. Besides this concern, the context of driving schools was also chosen for merely practical reasons: in a driving school, it is possible to modify and/or intervene in the lesson, e.g., by organizing small subgroups, without great difficulty, whereas it would be more difficult to do so in a school or university.

Anyhow, besides the specific case study, the present work aimed at comparing “in-presence” vs. “remote” modalities of teaching in order to point out the eventual differences in terms of students’ cognitive experiences, i.e., their mental states such as attention and workload during the lesson, and performance, i.e., the resulting learning effectiveness. It is important to underline that the overarching purpose of the study is not to determine which modality is better. On the contrary, considering the current scenario and new lifestyles, there is no doubt that online education is becoming a permanent practice, and therefore, we believe that obvious differences due to new educational methods should be evaluated in order to understand how to modify and make the most of them without losing efficiency. 

## 2. Materials and Methods

### 2.1. Participants

Forty (40) participants voluntarily took part in this experiment, receiving a gadget and a free practical driving lesson as compensation. They were all attending the course to obtain a Class B driving license. They were almost gender balanced (17 males and 23 females), and they were on average 25.9 (±11.6) years old. The sample size was the maximum achievable with the available resources. Anyhow, the sample size is in line with best practices and minimum requirements in the field of applied neuroscience [33].

All of the participants signed the informed consent form and the related information sheet, in which the study was explained, before participating in the experiment. In particular, in order to avoid any interference between the presence of monitoring devices and users’ behaviour, at the beginning, it was not stressed that biosignal acquisition was aimed at inferring information about their cognitive engagement during the lesson. They were told that the overarching aim was to validate those devices for their potential future use in this kind of application. At the end of the experiments, more information about the study was provided, and they were free to eventually withdraw their consent (but this did not happen). Authorisation to use the video/graphical material (i.e., photos and videos of the experiment) was also signed. The experiment was conducted following the principles outlined in the Declaration of Helsinki of 1975, as revised in 2008, and it received approval from the Sapienza University of Rome ethical committee (nr. 2507/2020).

### 2.2. Experimental Protocol

The participants had to follow a normal lesson of their driving license course, lasting 1 h and focused on the topic “Crossroads and related signage”. The 40 participants were divided into 5 different groups of 8 participants each due to some logistic concerns, mainly the difficulty of having too many devices streaming the collected data in the same room. Anyhow, the topic of the lesson was the same, and it was provided at the same time of day (4 p.m.) and by the same teacher in order to reproduce experimental conditions that were as similar as possible. Please refer to Figure 1 for the overall experimental design.

During each lesson, the 8 participants were divided into 2 subgroups of 4 people: 1 subgroup attended the lesson in person (namely, “In-presence condition”), while the other subgroup attended the same lesson remotely (namely, “Remote condition”). In the latter condition, the participants were placed in a separate room in the same driving school, and each one of them was seated at a personal workstation with a computer, a webcam and a couple of headphones with a microphone in order to allow interaction with the teacher; participants were connected by means of common teleconference software to the same lesson. After half an hour, i.e., after half the lesson, the two subgroups switched their positions, thus moving to the other condition. In this way, all of the participants experienced both conditions (“In-presence” and “Remote”), with half of them starting with the “In-presence” one and the other half starting with the “Remote” one.

During the whole lesson, neurophysiological data (please refer to Section 2.3) were recorded from all the participants. Before starting each lesson condition, an individual baseline was recorded for each participant. They were asked to remain relaxed for 1 minute in their seats in the classroom while looking at the teacher’s desk (“In-presence”) or at their workstation while looking at the computer with the teleconference software initialised (“Remote”).

At the end of the whole lesson, the participants had to fill out a multiple-choice questionnaire (which was the same for all of the participants) of 10 questions. The subjects of these questions were chosen in order to cover all of the topics of the lesson, with a balance between topics discussed during the first (5 questions) and the second (5 questions) halves of the lesson.

### 2.3. Data Collection

Neurophysiological data, in particular, ocular blinking (electrooculography, EOG), heart activity (photoplethysmography, PPG) and skin sweating (electrodermal activity, EDA), were recorded from all participants in both lesson conditions. At the end of the whole lesson, the participants had to fill out a questionnaire about the topics of the lesson. Below, data acquisition and analysis are described for each data type.

#### 2.3.1. EOG Signal Acquisition and Analysis

EOG signals were recorded by a commercial wearable device aimed at recording electroencephalographic (EEG) signals, i.e., Muse S (InteraXon Inc., Toronto, ON, Canada). It consists of a headband including four EEG channels placed over the prefrontal and temporal regions, namely, TP9, TP10, AF7 and AF8 of the 10-10 International System. The signals recorded from the four channels were collected with a sampling frequency of 256 Hz, and they are referred to as AFz. Due to the debated quality of EEG data collected by such a device [34,35], it was used only to estimate ocular blinking activity, with blinks having an intrinsically higher signal-to-noise ratio.

The whole data analysis was performed offline by means of Matlab (Mathworks Inc., Natick, MA, USA). Raw EEG data were digitally filtered by a 5th-order Butterworth band-pass filter (high-pass cut-off frequency = 1 Hz; low-pass cut-off frequency = 12 Hz). Such frequencies were chosen in order to consider the power spectrum range containing almost the entire spectral content of ocular blinks [36,37]. At this point, the channel (among the 4 available ones) with the most visible EOG pattern was selected by the same expert, in particular, a biomedical engineer with more than 4 years of experience. Then, the Reblinca algorithm [38] was used to detect eye blinks. In particular, the algorithm has been optimized by including three specific criteria based on the signal amplitude, shape and distribution in order to discriminate true (real blinks) from false (noise) positives. So, for each participant and for each condition, the Eye Blink Rate (EBR) was estimated as “Blinks per minute”. In particular, the inverse EBR was computed, since the EBR has been found to be inversely correlated with vigilance [39], as also described in the discussion. Then, the inverse EBR values of each subject were normalised by subtracting the individual mean baseline EBR value and dividing the result by the individual EBR standard deviation.

#### 2.3.2. PPG Signal Acquisition and Analysis

PPG signals were collected by means of a wearable bracelet, i.e., the Empatica E4 (Empatica Inc., Boston, MA, USA), worn on the wrist of the non-dominant hand. The signal was acquired with a sampling frequency of 64 Hz. 

The whole data analysis was performed offline by means of Matlab (Mathworks Inc., Natick, MA, USA). Raw PPG data were digitally filtered by using a 5th-order Butterworth band-pass filter (1–5 Hz) in order to exclude the continuous component, as well as slow signal drifting, and to emphasise the PPG signal patterns related to the pulse. At this point, the Pan–Tompkins algorithm [40] was employed to detect the pulse-related peaks so as to calculate the Inter-Beat Intervals (IBI signal). The so-obtained IBI signals were processed in order to remove any type of artefacts (such as the spurious oscillations visible in Figure 2, right side, in the PPG data) by means of the HRVAS Matlab suite [41]. At this point, clean IBI signals were processed to estimate the Heart Rate (HR) as “Beats per minute”. Then, the HR values of each subject were normalised by subtracting the individual mean baseline HR value and dividing the result by the HR individual standard deviation.

The IBI signal was also analysed to estimate the Heart Rate Variability (HRV). In particular, the HRV was analysed in the frequency domain by computing the Lomb–Scargle periodogram [42] of the IBI signal. Analysis has shown that the Lomb–Scargle periodogram can produce a more accurate estimate of the Power Spectrum Density (PSD) than Fast Fourier Transform methods for typical HR data. Since the HR data are unevenly sampled data, another advantage of the Lomb–Scargle method is that in contrast to Fast Fourier Transform–based methods, it is able to be used without the need to resample and detrend the RR data [43]. According to the scientific literature, the PSD of the HRV signal was computed over Low (LF: 0.04–0.15 Hz) and High Frequencies (HF: 0.15–0.4 Hz), and then the LF/HF ratio was computed as a relevant indicator of HRV [44]. The LF/HF values of each subject were normalised by subtracting the individual mean baseline LF/HF value and dividing the result by the individual HR standard deviation. 

#### 2.3.3. EDA Signal Acquisition and Analysis

EDA signals were collected by means of the previously mentioned wearable bracelet, i.e., the Empatica E4 (Empatica Inc., Boston, MA, USA), worn on the wrist of the non-dominant hand. The signals were acquired with a sampling frequency of 4 Hz.

The EDA was first low-pass-filtered with a cut-off frequency of 1 Hz, and then an artefact correction Matlab tool was applied in order to remove discontinuities and spurious peaks from the signals. Lastly, the signals were processed by using the Ledalab suite [R], a specific open-source toolbox implemented within the Matlab (MathWorks Inc., Natick, MA, USA) environment for EDA processing. A continuous decomposition analysis [45] was applied in order to estimate the tonic (SCL) and phasic (SCR) components. The SCL is the slow-changing component of the EDA signal, mostly related to the global arousal of the participant. On the contrary, the SCR is the fast-changing component of the EDA signal, usually related to single stimuli reactions. In this study, the SCR component estimation was affected by the low sampling frequency of the device (i.e., the Empatica E4); therefore, only the slow-varying SCL component was considered in the analysis. Additionally, in this case, the SCL values of each subject were normalised by subtracting the individual mean baseline SCL value and dividing the result by the SCL individual standard deviation.

#### 2.3.4. Questionnaire

At the end of the whole lesson, the participants had to individually fill out a multiple-choice questionnaire of 10 questions, the topics of which were balanced between the first (5 questions) and second (5 questions) halves of the lesson. The questions were the same for all participants and were taken from the official quiz used for the final exam.

#### 2.3.5. Statistical Data Analysis

All of the normalised neurophysiological parameters, i.e., the EBR, the HR, the HRV in terms of LF/HF and the SCL, were estimated by using the same 60-s time resolution, i.e., one value each minute. This time window length was chosen because it is the minimum requirement to obtain an adequately resolved power spectrum of the IBI signal in order to obtain a reliable HRV estimate. The other neuroparameters were coherently averaged. Then, each condition (lasting 30 min) was divided into ten 3-min time windows. For each condition, all of the neuroparameters were averaged across subjects for each time window; therefore, the result was a 10-point time series for each neuroparameter for both conditions (In-presence vs. Remote). The two conditions were then compared by means of paired statistical tests. In particular, a statistical non-parametric paired *t*-test, i.e., Wilcoxon signed-rank test, was employed to compare the two conditions since the data sample size was small (*n* = 10) and it was not possible to demonstrate the Gaussianity of data distributions. Coherently, the effect size was estimated by calculating the matched-pairs rank biserial correlation coefficient (analogous to Cohen’s d coefficient for non-parametric tests) [46].

Results from questionnaires were only qualitatively analysed.

## 3. Results

In the following, the results from 35 out of the 40 participants are presented, since 1 participant left the experiment before its conclusion, while 4 participants had at least one corrupted data point, so only participants having a fully sound dataset were included in the analysis.

### 3.1. Neurophysiological Parameters

The analysis of the inverse Eye Blink Rate parameter (Figure 3) showed that the EBR was almost significantly higher (lower values when considering the inverse) in the *Remote* condition than in the *In-presence* condition (Wilcoxon signed-rank test: *z* = 1.886; *p* = 0.064; effect size = 0.673).

In particular, the difference was more evident after the first 15 min, where the inverse Eye Blink Rate in the *Remote* condition always remained lower than that in the *In*-*presence* one, and even lower than the “0” level (Figure 3b). Because the scores are normalised with respect to the baseline, this means that in the second part of the lesson in the remote condition, the students experienced a blink frequency even higher (thus, with lower inverse values) than during the initial resting condition, i.e., the baseline.

In terms of heart activity, a significant effect was found both in terms of the Heart Rate and its Variability. In particular, the analysis of the Heart Rate parameter (Figure 4) showed significantly higher values in the *In-presence* condition than in the *Remote* condition (Wilcoxon signed-rank test: *z* = 2.497; *p* = 0.016; effect size = 0.891).

On the other hand, the analysis of the Heart Rate Variability (Figure 5) showed significantly higher LF/HF values in the *In-presence* condition than in the *Remote* condition (Wilcoxon signed-rank test: *z* = 2.396; *p* = 0.014; effect size = 0.855).

With regard to skin sweating, the analysis of the Skin Conductance Level (Figure 6) showed significantly higher values in the *Remote* condition than in the *In-presence* condition (Wilcoxon signed-rank test: *z* = −2.599; *p* = 0.006; effect size = −0.927).

### 3.2. Questionnaires

The results of the questionnaires were analysed from a qualitative point of view. First, for each participant, the questions were categorised into subjects covered in the In-presence and Remote conditions. Then, the percentage of right and wrong answers were calculated for each condition (Figure 7). The results showed an increase in errors in the Remote condition, where subjects gave 3.5% more wrong answers (15.2% vs. 11.7% wrong answers in the In-presence condition).

Since these results could have been biased by only a few students providing a large number of wrong answers in the Remote condition, for each student, the number of wrong answers was compared in both conditions, and the student’s performance was accordingly assessed as follows (Figure 8):“Worst by PRESENCE”, if the participant gave more wrong answers for subjects covered in the *In-presence* condition;“Worst by REMOTE”, if the participant gave more wrong answers for subjects covered in the *Remote* condition;“EQUAL”, if the number of wrong answers was the same between the two conditions.

Half of the sample (55.17%) did not show any difference, and almost one-third of them (31.03%) had a worse performance in the Remote condition, while only 13.79% had a worse performance in the Presence condition.

## 4. Discussion

The present study involved forty students at a driving school, in a real classroom, in order to evaluate and compare their experiences in face-to-face and remote learning conditions. Wearable devices to measure EEG, PPG and EDA signals were employed in order to analyse the neurophysiological activities of the students during both modalities and therefore to obtain insights into the cognitive dimension. In particular, four different neuroindicators were considered: the Eye Blink Rate (estimated from the EOG component of the EEG data), the Heart Rate and its Variability (estimated from the PPG data) and the Skin Conductance Level (estimated from the EDA data). Additionally, the students filled out a questionnaire at the end in order to obtain an explicit measure of their learning performance.

The Eye Blink Rate (EBR) was investigated as a measure of attention. In fact, the EBR has been demonstrated to be inversely correlated with attention and vigilance: i.e., an increased EBR is a biomarker of decreased attention, the loss of situation awareness and even drowsiness [39,47,48]. Such findings have also been demonstrated in the scientific literature related to education and learning [49,50]. In the present work, for convenience, the inverse of the EBR was considered in order to have a direct correlation with attention. An almost significant effect was found (Figure 3), since the EBR tended to be lower (and so attention was higher) in the *In-presence* condition. Actually, statistical significance was not achieved (*p* = 0.064), but the effect size (=0.673) was between medium (Cohen’s *d* = 0.5) and large (Cohen’s *d* ≥ 0.8) [45]. The difference was more evident after the first 15 min, where the inverse Eye Blink Rate in the *Remote* condition, i.e., the attention level, always remained lower than that in the *In-presence* one and even lower than the “0” level (Figure 3b), i.e., the baseline.

In terms of heart activity, a significant effect was found in terms of both the Heart Rate (*p* = 0.016) and Heart Rate Variability (*p* = 0.014), with a large effect size (>0.8) in both cases. HR has been demonstrated to be positively correlated with Mental Workload, i.e., an increase in this indicator should suggest higher mental effort [51,52]. The HRV indicator, computed as the ratio between Low and High Frequencies, is considered a biomarker of attention and mental effort. This indicator has been demonstrated to increase when the user is cognitively involved in a task [47,53]. In other words, both parameters are positively correlated with cognitive engagement and mental resource allocation, as also demonstrated in previous works [54,55,56]. In the present work, both parameters were greater for the *In-presence* condition, therefore suggesting higher cognitive engagement than in the *Remote* condition. Anyhow, it is important to point out that in both conditions, the values were positive, i.e., higher than baseline, and thus, the students were less but still mentally engaged in the *Remote* lesson as well. It is also interesting to note that, in terms of HR, the difference between the two conditions appears after 18 min (Figure 4b).

Last but not least, from a neurophysiological point of view, the analysis of skin sweating also pointed to a significant effect. In fact, the Skin Conductance Level was significantly higher in the *Remote* condition and, only after 24 min, converged to values similar to those measured in the *In-presence* condition. The SCL is usually considered a biomarker of physiological arousal and even stress [57,58,59]. Previous works in the education field demonstrated a positive correlation between skin conductance, in particular, when it increases over baseline values, and students’ stress and aggressive behaviours [60,61]. In the present works, while no evident SCL fluctuations were found in the *In-presence* setting, a large increase was measured at the beginning of the *Remote* condition. This sign of stress could be explained by the specific “remote modality” considered in this study, i.e., an interactive remote modality in a hybrid classroom with people also present. In this way, such physiological effects could be due to certain discomfort, and even stress, experienced because of the less immediate interaction with the teacher remotely. Technical or functional issues, such as speaking with the microphone still muted or difficulties participating in the discussion because the teacher is distracted by other people present, could cause stress and thus lead to loss of interest and attention. Interestingly, this effect tends to disappear at the end, when the other indicators suggested decreases in attention and mental engagement. In other words, it seems that in a hybrid classroom, sustaining attention for a prolonged time and remaining mentally engaged are more demanding and stressful.

Not surprisingly, the analysis of questionnaires, intended as a measure of “learning performance”, showed an increase in errors on topics learned remotely, with performance deteriorating in about one-third of the students.

In any case, as mentioned in the introduction, the overarching aim of the study was not to determine which modality is better. On the contrary, considering the current scenario and new lifestyles, the rationale was to deploy innovative methodologies for investigating and pointing out differences, in terms of students’ experiences, between different educational modalities in order to define strategies and countermeasures to deal with raised concerns. As a practical example, from this study, it could be concluded that in hybrid modalities, the interaction of people learning remotely should be facilitated and encouraged, and the lesson should be organised in more and shorter modules, for instance, four 15-minute-long modules instead of two 30-minute-long modules, in order to avoid decreases in attention and engagement. 

To the best of our knowledge, this is the first study simultaneously employing different neuro-devices investigating both the Central (brain activity) and Autonomic (heart and sweating activities) Nervous Systems, on a non-negligible sample size (i.e., 40 participants), in a real setting and considering both educational modalities, i.e., *In-presence* and *Remote* conditions. 

The study is, however, intended to be preliminary due to the novelty and complexity of the experimental approach, and therefore, it is limited by some shortcomings, such as the lack of valid electroencephalographic data, despite the use of a wearable EEG device, and the loss of 5 out of the 40 participants, thus affecting the data amount and distributions. In future studies, a similar approach should be enriched by a deeper analysis of students’ brain activity, should involve a larger sample size, and should be applied in different contexts (besides driving training) in order to demonstrate the cross-domain validity of the obtained outcomes. Additionally, on a more applied level, it would be interesting to compare different alternatives of remote modalities, for example, a fully remote classroom (instead of a hybrid situation such as that investigated) or remote modalities without interactions with the teacher (i.e., pre-recorded courses). Last but not least, the proposed neurophysiological approach could be evaluated with respect to, and eventually combined with, recent tools aimed at assessing students’ learning performance such as Learning Analytics [62,63]. 

Anyhow, beyond the results obtained and discussed, this study suggests how the use of these new wearable technologies opens new scenarios for the study of students' experiences to improve the effectiveness of teaching and optimise the use of new educational modalities that are becoming widespread.

## 5. Conclusions

The analysis of neurophysiological indicators highlights higher “cognitive activity”, in terms of attention and mental engagement, during the *In-presence* lesson compared to the *Remote* modality. On the other hand, the analysis of skin sweating seems to suggest higher stress, particularly during the first part of the lesson, in students in the *Remote* setting. This could be due to the less smooth interaction with the teacher. The analysis of questionnaires demonstrated worse performance on questions related to the subjects taught remotely. Therefore, there is evidence of a common “disengaging” behaviour when attending remote courses, which could undermine teaching effectiveness.

In general, the use of physiological indicators could help to obtain insights into possible mental causes, which are often “blind” to an external supervisor, such as decreasing attention and increasing stress, as well as their dynamics during the lesson itself. Neuroscientific tools have become a powerful means that could add value to traditional assessment methods and could allow the definition of proper countermeasures to emerging issues when living through periods of transition to new tools, practices and protocols.

## Figures and Tables

**Figure 1 brainsci-13-00095-f001:**
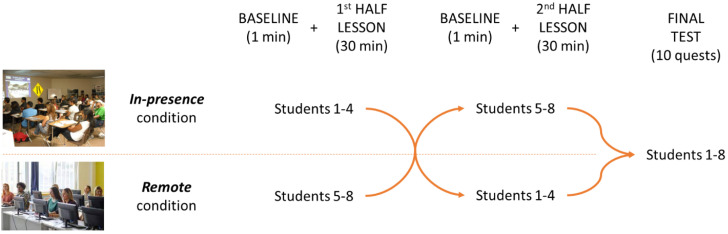
Overview of the experimental protocol for a subgroup of 8 participants.

**Figure 2 brainsci-13-00095-f002:**
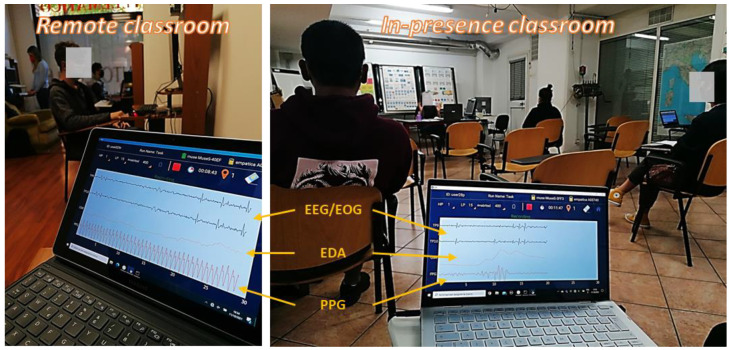
Two pictures of signal recording during the experimental task in both conditions (IN-PRESENCE and REMOTE). In particular, the first two blue channels are related to the EEG data gathered from Muse S, from which the EOG pattern was obtained (the blink shapes are visible). The last two orange channels are, respectively, the EDA and PPG data gathered from Empatica.

**Figure 3 brainsci-13-00095-f003:**
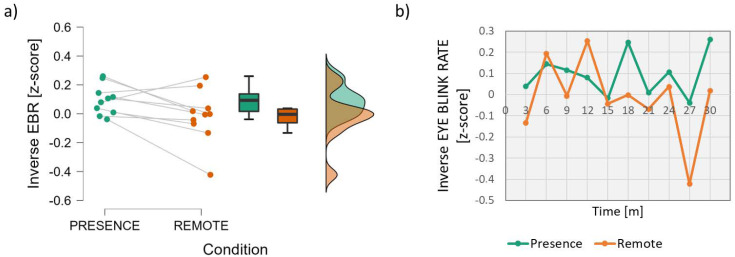
Analysis of the inverse Eye Blink Rate parameter. On the left (**a**), mean and confidence interval (95%) of the distributions related to the two conditions (In-presence and Remote). On the right (**b**), the dynamics of the inverse EBR averaged across students during the time span of the lesson.

**Figure 4 brainsci-13-00095-f004:**
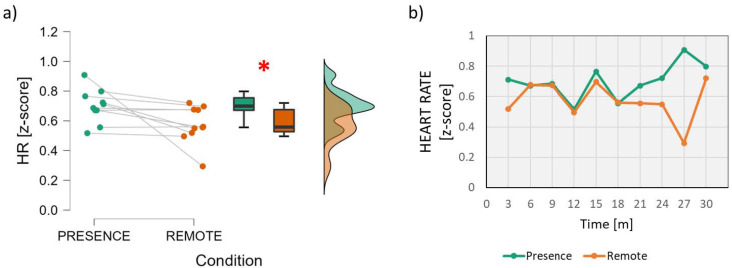
Analysis of the Heart Rate parameter. On the left (**a**), mean and confidence interval (95%) of the distributions related to the two conditions (In-presence and Remote). The red asterisk indicates the presence of a statistically significant effect. On the right (**b**), the dynamics of the HR averaged across students during the time span of the lesson. The red asterisk indicates the statistically significant (*p* < 0.05) result.

**Figure 5 brainsci-13-00095-f005:**
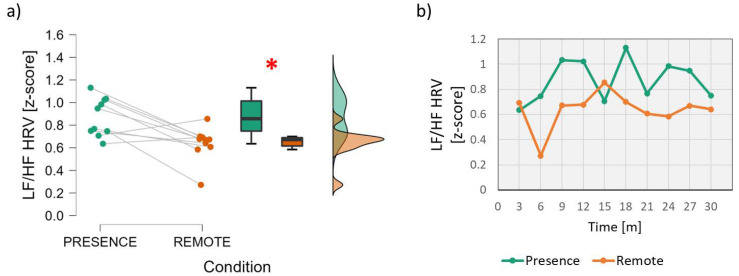
Analysis of the Heart Rate Variability, particularly the ratio between Low and High Frequencies. On the left (**a**), mean and confidence interval (95%) of the distributions related to the two conditions (In-presence and Remote). The red asterisk indicates the presence of a statistically significant effect. On the right (**b**), the dynamics of the HRV parameter averaged across students during the time span of the lesson. The red asterisk indicates the statistically significant (*p* < 0.05) result.

**Figure 6 brainsci-13-00095-f006:**
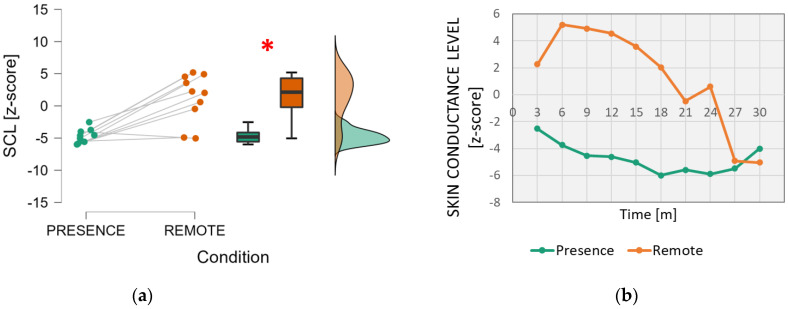
Analysis of the electrodermal activity, particularly the Skin Conductance Level. On the left (**a**), mean and confidence interval (95%) of the distributions related to the two conditions (In-presence and Remote). The red asterisk indicates the presence of a statistically significant effect. On the right (**b**), the dynamics of the SCL parameter averaged across students during the time span of the lesson. The red asterisk indicates the statistically significant (*p* < 0.05) result.

**Figure 7 brainsci-13-00095-f007:**
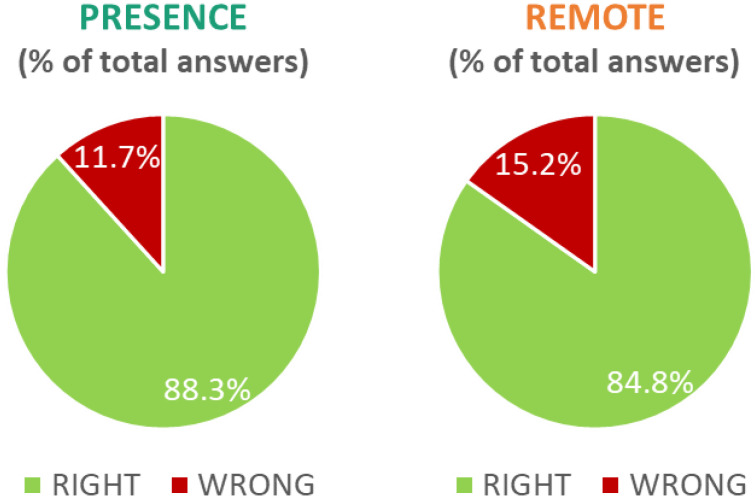
Percentage of right and wrong answers provided by the participants to the subjects covered in the In-presence (**left**) and Remote (**right**) conditions.

**Figure 8 brainsci-13-00095-f008:**
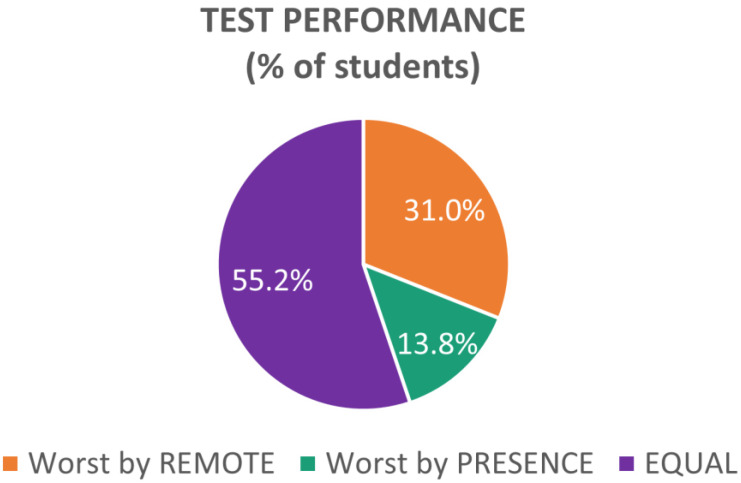
Percentage of students divided according to their test performance: in orange, students who gave more wrong answers in the Remote condition; in green, students who gave more wrong answers in the Presence condition; in violet, students who had the same number of errors between the two conditions.

## Data Availability

The aggregated data analysed in this study might be available on request from the corresponding author.

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
