# Peer review of "Neurophysiological Evaluation of Students’ Experience during Remote and Face-to-Face Lessons: A Case Study at Driving School"

_brainsci, 2023, doi:10.3390/brainsci13010095_

Round 1

Reviewer 1 Report

The article presents an interesting case study on a neurophysiological evaluation of students’ experience at a driving school during remote and face-to-face lessons. It is a very interesting article.

Comments and suggestions

·         Minor format issues

Abstract. - remove space between lines 18-19; 25-26; 30-31, Or even writing the whole abstract in a single paragraph.

In-presence (line 147); IN-PRESENCE (previous to line 163; in the text related to Figure 2); In Presence (line 275 and line 286); In presence (previous to line 280; in the text related to Figure 5).

The same that happens with In-presence happens with Remote.

Line 226.- ‘…estimate. the tonic...’ -> delete the point

Additional considerations

Optional. - Some additional references about engagement in online formats could be included to enrich the introductions section.

Future lines are included in the Discussion section. However, limitations of the research are not mentioned.  

References

Some doi are missing, e.g.

[7] S. Unger e W. R. Meiran, «Student Attitudes towards Online Education during the COVID-19 Viral Outbreak of 449 2020: Distance Learning in a Time of Social Distance», Int. J. Technol. Educ. Sci., vol. 4, fasc. 4, pp. 256–266, 2020.

DOI:10.46328/ijtes.v4i4.107

Author Response

We would like to thank the reviewer for the remarks and the useful suggestions, that helped us to improve the manuscript. Here attached a file with our answers listed point by point.

Best regards,

Gianluca and colleagues

Reviewer 2 Report

Please refer to the document attached.

Best regards

Thank you

Author Response

(The authors gave the same response as above.)

Reviewer 3 Report

Dear authors,

Congratulations on your manuscript.

It is very well written regarding content and results.

Nevertheless, I have some suggestions:

1. keywords: I would add words such as face-to-face, remote, neurophysiological approach, learning performance, instead of neuroscience, education, learning, and neuroimaging;

2. in the introduction section it is clear what is the aim but I lack specific objectives such as for instance, "students' cognitive experience" and their "performance";

3. in the methods section please explain the choice of the driving school and the sample (which is small);

4. Please detail and explain the limitations of the study in the discussion section including the lack of gaussianity distribution as the number of valid answers (n=35);

5. In future studies, in the conclusions section, it would be very interesting to investigate and test the effectiveness of the different modalities regarding students´ cognitive experience and their learning performance.

Compliments,

Author Response

(The authors gave the same response as above.)
